# Harnessing Attenuation-Related Mutations of Viral Genomes: Development of a Serological Assay to Differentiate between Capripoxvirus-Infected and -Vaccinated Animals

**DOI:** 10.3390/v15122318

**Published:** 2023-11-25

**Authors:** Francisco J. Berguido, Tesfaye Rufael Chibssa, Angelika Loitsch, Yang Liu, Kiril Krstevski, Igor Djadjovski, Eeva Tuppurainen, Tamaš Petrović, Dejan Vidanović, Philippe Caufour, Tirumala Bharani K. Settypalli, Clemens Grünwald-Gruber, Reingard Grabherr, Adama Diallo, Giovanni Cattoli, Charles Euloge Lamien

**Affiliations:** 1Animal Production and Health Laboratory, Animal Production and Health Section, Joint FAO/IAEA Division, Department of Nuclear Sciences and Applications, International Atomic Energy Agency, WagramerStrasse 5, P.O. Box 100, 1400 Vienna, Austria; 2Institute of Biotechnology, University of Natural Resources and Life Sciences (BOKU), Muthgasse 18, 1190 Vienna, Austria; 3Animal Health Institute, Sebeta P.O. Box 4, Ethiopia; 4Austrian Agency for Health and Food Safety (AGES), Spargelfeldstrasse 191, 1220 Vienna, Austria; 5China National Clinical Research Center for Neurological Diseases, Beijing Tiantan Hospital, Capital Medical University, Beijing 100070, China; 6Faculty of Veterinary Medicine, Ss. Cyril and Methodius University in Skopje, 1000 Skopje, North Macedonia; 7Institute of International Animal Health/One Health, Friedrich-Loeffler-Institut, 17493 Greifswald, Germany; 8Scientific Veterinary Institute “Novi Sad”, 21000 Novi Sad, Serbia; 9Veterinary Specialized Institute Kraljevo, Zicka 34, 36103 Kraljevo, Serbia; 10UMR ASTRE Cirad-Inrae, University of Montpellier (I-MUSE), 34398 Montpellier, France; 11Core Facility Mass Spectrometry, University of Natural Resources and Life Sciences Vienna, 1190 Vienna, Austria; 12Independent Researcher, Hahngasse, 24-26, 02/07, 1090 Vienna, Austria

**Keywords:** capripoxvirus, iELISA, B22R, DIVA, LSDV, SPPV, GTPV, serology, neethling, KSGP 0240 (KS-1) vaccine

## Abstract

Sheeppox, goatpox, and lumpy skin disease caused by the sheeppox virus (SPPV), goatpox virus (GTPV), and lumpy skin disease virus (LSDV), respectively, are diseases that affect millions of ruminants and many low-income households in endemic countries, leading to great economic losses for the ruminant industry. The three viruses are members of the *Capripoxvirus* genus of the *Poxviridae* family. Live attenuated vaccines remain the only efficient means for controlling capripox diseases. However, serological tools have not been available to differentiate infected from vaccinated animals (DIVA), though crucial for proper disease surveillance, control, and eradication efforts. We analysed the sequences of variola virus B22R homologue gene for SPPV, GTPV, and LSDV and observed significant differences between field and vaccine strains in all three capripoxvirus species, resulting in the truncation and absence of the B22R protein in major vaccines within each of the viral species. We selected and expressed a protein fragment present in wildtype viruses but absent in selected vaccine strains of all three species, taking advantage of these alterations in the B22R gene. An indirect ELISA (iELISA) developed using this protein fragment was evaluated on well-characterized sera from vaccinated, naturally and experimentally infected, and negative cattle and sheep. The developed wildtype-specific capripox DIVA iELISA showed >99% sensitivity and specificity for serum collected from animals infected with the wildtype virus. To the best of our knowledge, this is the first wildtype-specific, DIVA-capable iELISA for poxvirus diseases exploiting changes in nucleotide sequence alterations in vaccine strains.

## 1. Introduction

SPPV, GTPV, and LSDV are large double-stranded DNA viruses of the genus *Capripoxvirus* within the family *Poxviridae* [1].

SPPV and GTPV infect sheep and goats, causing sheeppox (SPP) and goatpox (GTP), and can also infect wildlife [2,3].

LSDV causes lumpy skin disease (LSD) in cattle and buffalo, though there are reports of LSDV infection in Arabian oryx, Springbok antelope, Oryx gazelle, Asian water buffalo, and Eland [4,5,6].

Due to their economic impact on the cattle, sheep, and goat farming industries and their potential for rapid transboundary spread, LSD, SPP, and GTP are categorized by the World Organisation for Animal Health (WOAH) as notifiable diseases [7]. They are also listed as potential bioterrorist agents by the United Kingdom and the United States Department of Agriculture [8,9].

SPP and GTP are endemic in Asia, Africa (except for southern Africa), the Middle East, and Turkey [10,11]. In addition, several sporadic SPP incursions have occurred in Greece and some Eastern European countries since 2013 [10].

LSD, on the other hand, was endemic only in the Middle East, sub-Saharan Africa, and Egypt [12] until August 2015, when the first outbreaks occurred in Europe (Greece), followed by additional outbreaks in September of the same year in the Russian Federation and Kazakhstan [10,13]. By 2016, LSD was present in Bulgaria, the Republic of North Macedonia, Serbia, Montenegro, Albania, and Kosovo in Europe [14,15]. In 2019, the disease emerged in China, India, and Bangladesh in Asia [16,17,18]. By March 2022, the disease had spread to Malaysia, Cambodia, Mongolia, Pakistan, the Lao People’s Democratic Republic, and Singapore [19].

Vaccination remains the best way to protect animals and control the spread of capripox [14,20]. Currently, LSD vaccines are mostly live-attenuated homologous vaccines, such as those based on the Neethling and KSGP 0240 (also known as KS-1) vaccine strains [17,18]. In addition to LSDV vaccines, higher doses of SPPV, as well as GTPV live-attenuated vaccines, are used against LSD [14,20,21,22]. For SPP and GTP, live-attenuated vaccines such as those based on the SPPV Yugoslavian RM65, SPPV Romanian, and GTPV Gorgan vaccine strains are commonly used [14].

One of the challenges in using live-attenuated vaccines is the maintenance of the seed viruses so that they do not become too attenuated and lose their ability to generate a proper immune response or become too virulent to cause infection [20].

Vaccines have also been the cause of farmers’ concerns during vaccination campaigns, mainly because they could induce disease-related symptoms as side effects [23,24].

Due to low dosages, poor attenuation levels, or partial protection, both heterologous and homologous live-attenuated vaccines can become sources of disease spread [25,26,27,28], calling for the urgent need to develop tools that can differentiate vaccinated animals from infected ones when outbreaks occur in vaccinated herds.

To address this issue, several tests were developed for molecular identification and differentiation of capripoxviruses, including some that can differentiate vaccine strains from field strains [29,30,31]. However, a serological DIVA (Differentiating between Infected and Vaccinated Animals) test for capripox has been, so far, unavailable. Therefore, the objective of this study was to analyse and evaluate the Variola virus (VARV) B22 homologue for capripoxviruses as a candidate target to develop an iELISA capable of differentiating between vaccinated and naturally infected animals. The goal of this assay is to provide the means for effective serological surveillance campaigns, epidemiological studies, and countries’ declaration of freedom from disease and potentially be the basis for future capripox eradication efforts [20].

## 2. Materials and Methods

### 2.1. Target Design

In our previous work, the comparison of the genomes of SPPVs, sheeppox virus 17077–99 (AY077832), sheeppox virus A (AY077833), and sheeppox virus NISKHI (AY077834), retrieved from GenBank, and the unpublished genome of the Romanian vaccine strain used in Morocco, revealed regions with unique differences between SPPV vaccines and SPPV field isolates within the VARV B22R homologue gene of SPPV [32]. These attenuation markers consist of a series of deletions in the B22R gene in the vaccine strains, leading to the early termination of the protein. Similar comparative analyses using the entire genome of vaccines and field strains of LSDV and GTPVs available in public databases, in our laboratories, and by earlier reports suggested a similar alteration in vaccine strains of each capripoxvirus genotype [33].

Based on these observations, we selected a region in sheeppox virus 10700–99 strain TU-V02127 (AY077832-aa 784 to 929), which would only be expressed in virulent SPPV and not in vaccines derived from Romanian and Yugoslavian RM65. The analysis of other capripoxviruses suggested that the corresponding fragment would also be present in LSDV and GTPV field viruses but not in LSDV vaccines derived from the Neethling strain or GTPV vaccines derived from GTPV G20-LKV.

### 2.2. Sequence Analysis, Protein Domain Prediction and Graphic Illustration

Nucleotide and amino acid sequence alignments were performed using BioEdit (v7.2.6) sequence alignment editor. Protein prediction was performed using the TMSEG method implemented in the PredictProtein software [34]. Graphic illustrations were created by BioRender.com.

### 2.3. Protein Expression

The designed B22R gene fragment was codon-optimized, and the synthetic gene was cloned into the pET28a+ expression vector (Twist Biosciences, South San Francisco, CA, USA) and expressed as described below. Induction was performed in BL21 (DE3) cells with 0.1 mM of IPTG and 0.3% L-arabinose. After cellular lysis, the supernatants were harvested by centrifugation at 6000× *g* for 20 min and purified using fast-flow crude his-trap columns (Cytiva, MA, USA) according to the manufacturer’s instructions. Additional purification was performed by size exclusion chromatography, Superdex Increase 75 10/300 (Cytiva, MA, USA). All the purification steps were performed with an ÄKTA-Go protein purification system (Cytiva, MA, USA). The expression of the protein was confirmed by SDS-PAGE and Western blot using an HRP-conjugated anti-his-tag antibody (dilution 1 in 1000; Merck KGaA, Darmstadt, Germany) and undiluted hyperimmune sheeppox positive serum [35]. Reactive bands were visualized using the ECL Prime Western-blot-Detection reagent (Cytiva, MA, USA) and detected by the Molecular Imager Gel Doc XR System (BioRad, CA, USA). 

### 2.4. Peptide Mapping by Mass Spectroscopy

The sequence of the purified protein fragment was verified by mass spectroscopy at the Core Facility for Mass Spectrometry at the University of Natural Resources and Life Sciences, Vienna, Austria. The sample was digested in solution. The proteins were S-alkylated with iodoacetamide and digested with Trypsin (Promega, Madison, WI, United States). Detection was performed with an iontrap MS (amazon speed ETD, Bruker) equipped with the standard ESI source in positive ion, DDA mode (= switching to MSMS mode for eluting peaks). MS scans were recorded (range: 150–2200 Da), and the 8 highest peaks were selected for fragmentation. The files were searched against an *E. coli* database (downloaded from Uniprot).

### 2.5. Reference Serum Samples Used

This study used six serum types: (a) Bovine LSD positive sera from either experimental (from the UK) or field-infected (from Bulgaria or the Republic of North Macedonia) animals (*n* = 32). (These samples were confirmed positive by serum neutralization test (SNT) or double antigen ELISA). (b) Bovine LSD-positive sera from vaccinated animals from Serbia (*n* = 75) was also used. The sera were collected after one year of a single round of Neethling vaccination or eight weeks after two rounds of yearly Neethling immunizations. These samples were confirmed positive by SNT or double antigen ELISA (Appendix A). (c) Bovine LSD negative sera (*n* = 73) from countries free of the infection, namely, France (provided by IDVet, Montpellier, France), Austria, and the Republic of North Macedonia (collected before 2010) was also used; (d) capripox-positive sheep and goat sera (*n* = 27) from animals experimentally infected with a virulent strain were also used. For SPPV, the sera were from three Malian sheep experimentally infected with SPPV Djelfa [35]. For GTPV, the sera were from four Ethiopian goats experimentally infected with GTPV Oman 84 [36]. These sera were confirmed positive by SNT; (e) sheeppox Romanian and LSDV KSGP 0240 (KS-1) vaccine sheep sera samples, experimentally infected collected at >15 days post-infection (DPI) (*n* = 8) from Pirbright, UK and AGES, Austria, and (f) capripox-negative sheep and goat sera (*n* = 87) from Austria, where the disease is not present.

### 2.6. Additional Sera

#### 2.6.1. Specificity (Parapox-Positive) Sera

Twelve goat samples infected with ORF (contagious pustular dermatitis virus) and nine pseudocowpox cattle sera from Zambia (kindly provided by Maureen Ziba—CVRI) were used to confirm the specificity of the capripox wildtype specific (wts)-DIVA iELISA.

#### 2.6.2. Longitudinal Sera

Two sets of samples from longitudinal studies were used. The first set comprised samples collected at days 0, 6, 12, 18, 20, 23, 26, and 30 DPI from two cattle experimentally infected with virulent South African LSDV Neethling strain [37]. The second set consisted of samples collected at 0, 7, 14, 21, 28, 35, 42, 49, and 56 DPI from two goats that were experimentally infected with GTPV Oman 84. This was part of a larger SPPV/GTPV study from experimentally infected Ethiopian goats [36].

#### 2.6.3. LSD-Positive Control Sera

Sera samples from cattle experimentally infected with virulent South African LSDV Neethling strain and confirmed positive by double antigen ELISA were pooled and used as a control [37].

#### 2.6.4. GTP/SPP-Positive Control Sera

Sera samples from Ethiopian goats experimentally infected with GTPV Oman 84 and confirmed positive by double antigen ELISA were pooled and used as control [36].

#### 2.6.5. Samples from the 2016 LSD Outbreak in the Republic of North Macedonia

Forty-five serum samples were collected from cattle vaccinated during the 2016 LSD outbreak in the Republic of North Macedonia. These animals had presented LSD symptoms >10 days after the vaccination when sera and nasal swaps were collected.

#### 2.6.6. Serum Neutralization Test

In order to determine sera positivity, SNT was performed under BSL-3 conditions in the laboratories of the Austrian Agency for Health and Food Safety (AGES, Austria). The general SNT is described as follows. Briefly, in quadruplicate, heat-decomplemented (56 °C for 1 h) sera samples and positive and negative controls were two-fold diluted (1 in 8 to 1 in 1280) in DMEM cell culture medium (Thermo Fisher Scientific, MA, USA) without serum or antibiotics and incubated with and without 100 TCID_50_ final concentration of wildtype LSDV Massalamia P4 (kindly provided by the Central Veterinary Research Laboratory, Sudan) for one hour at room temperature. Then, 20,000/well ESH-L cells (kindly provided by AGES, Austria) were added. The plates were incubated at 37 °C for 5 to 7 days, after which they were examined for cytopathic effects (CPE). A sample where CPE was observed in three or four wells out of four was considered negative; two out of four was considered borderline, and a sample where CPE was observed in zero to one well was considered positive. The final sera titre was determined as the highest dilution, where the samples in quadruplicate tested negative.

#### 2.6.7. Capripox IDVet Double Antigen ELISA

The capripox IDVet double antigen ELISA was used following the manufacturer’s instructions.

#### 2.6.8. Capripox Wts-DIVA iELISA

The best coating conditions for the assay were determined via a checkerboard dilution study. Additionally, the optimization of blocking buffer and secondary antibody conditions was based on our previous A34 capripox ELISA [38]. The capripox wts-DIVA iELISA tests on non-inactivated samples were performed under BSL-3 conditions (AGES, Austria). For the capripox wts-DIVA iELISA, polysorb plates (NUNC, Roskilde, Denmark) were coated overnight at 4 °C with 60 ng/well of the above-described fragment of the VARV B22R homologue gene product in 0.1 M carbonate buffer (pH 9.6). Although the antigen used and coating conditions of the plates were the same for LSD and SPP/GTP wts-DIVA iELISAs, we used different sera and secondary antibody dilutions per species. When the serum was of bovine origin, the conditions followed the LSD wts-DIVA iELISA described below. When the serum was of sheep or goat origin, the conditions followed the SPP/GTP wts-DIVA iELISA described below.

#### 2.6.9. LSD Wts-DIVA iELISA

Wells selected for testing were incubated with 50 uL/well of blocking buffer (Thermo Fisher Scientific, MA, USA) for 30 min at 37 °C. After discarding the buffer, 100 µL/well of serum samples and positive and negative controls previously diluted 1 in 100 in blocking buffer were incubated on the plates at 37 °C for 90 min. Then, the wells were washed 3 × 250 µL/well with 1× PBS plus 0.05% Tween 20 (PBS-T) and incubated for 45 min at room temperature with 100 uL/well of secondary antibody (Merck KGaA, Darmstadt, Germany) previously diluted 1 in 30,000 in blocking buffer. The wells were washed with PBS-T as described above, and 100 µL/well of TMB substrate (Mabtech, Nacka Strand, Sweden) were added. After 12 min in the dark, 100 µL/well of 1 M phosphoric acid (Merck KGaA, Darmstadt, Germany) was added to stop the reaction. Plates were read at 450 nm with the Multiskan Go ELISA plate reader (Thermo Fisher Scientific, MA, USA). The absorbance was calculated by subtracting the optical density (OD) measured on wells without sera from the values obtained from the sera-containing wells for each sample.

#### 2.6.10. SPP/GTP Wts-DIVA iELISA

Wells selected for testing were incubated with 50 µL/well of blocking buffer (Thermo Fisher Scientific, MA, USA) for 30 min at 37 °C. After discarding the buffer, 100 µL/well of serum samples and positive and negative controls previously diluted 1 in 400 in blocking buffer were incubated on the plates at 37 °C for 90 min. Then, the wells were washed 3 × 250 µL/well with PBS-T and incubated for 45 min at room temperature with 100 µL/well of secondary antibody (Merck KGaA, Darmstadt, Germany) previously diluted 1 in 20,000 in blocking buffer. The wells were washed with PBS-T as described above, and 100 µL/well of TMB substrate (Mabtech, Nacka Strand, Sweden) were added. After 12 min in the dark, 100 µL/well of 1 M phosphoric acid (Merck KGaA, Darmstadt, Germany) was added to stop the reaction. Plates were read at 450 nm with the Multiskan Go ELISA plate reader (Thermo Fisher Scientific, MA, USA).

The absorbance was calculated by subtracting the optical density (OD) measured on wells without sera from the values obtained from the sera-containing wells for each sample.

### 2.7. Sequencing

Fragments containing the partial sequences of the capripoxvirus homologue of the VARV B22R were sequenced in samples collected from the 2016 LSD outbreak in vaccinated herds in North Macedonia. Amplification was described earlier [39] and sequenced at LGC genomics (Berlin, Germany). The raw sequences were assembled using the Vector NTI software (Invitrogen) version 11.5, and the sequences were aligned using BioEdit version 7.2.5.

### 2.8. Statistical Analysis

The iELISA raw OD values, relevant information about the samples, and the SNT results were compiled in Microsoft Excel. The background-subtracted OD values and the S/P% values ([Raw OD of sample/Raw OD Pos] × 100) were calculated in Microsoft Excel, and these data were imported into R for further analysis. In addition to R-based functions, the dplyr package and tidyr package were used for data frame manipulation and statistical analysis. The pROC package [40] was used for the ROC analysis, and ggplot2 package [41] was used for the graphical representation of the data. Three methods, Youden’s index = sensitivity + specificity − 1 [42]; the Euclidean index = (1 − sensitivity)^2^ + (1-specificity)^2^ [43], and the product index = sensitivity * specificity [44], were used to determine the cut-off from the ROC analysis. The maximum values for Youden’s index, the product index, and the minimum value for Euclidian’s index were used as a criterion for selecting the optimal cut-off point.

## 3. Results

### 3.1. Target Sequence and Domain Prediction

The sequence alignment and analysis of several field and vaccine strains of the SPPV homologue gene of the VARV B22R showed a series of deletions in the vaccine but not in the field strains, including a 14-nucleotide deletion starting at position 2349 of the field reference strain (AY077832) (Figure 1). For vaccine strains of LSDV and GTPV, there was a single nucleotide (A) insertion at position 2148 and 2107, respectively. Any of these events resulted in a frameshift (Figure 1). On an amino acid level, the change translated into a truncated protein (stop codon shortly after the mutations).

The protein domain predictions showed that the DIVA ELISA target was present and exposed in the extracellular domain of cells infected with SPPV, LSDV, and GTPV field strain. In contrast, the DIVA target peptide was truncated and absent in the described vaccine strains of SPPV, LSDV, and GTPV (Figure 2 and Appendix A). 

We, therefore, selected and expressed a fragment of the wildtype strain after the frameshifts of SPPV, GTPV, and LSDV that would not be expressed by the vaccine strains: 

MDSRFASSICNARGLDLANYRGDSNIYKTTDDDFVKRENSLYARSKLEPELKDNPLYESLSNIDIVSNPYNNPKLSRRNAIKKKVLNDGYDEFVIRTDEEPSEKPNMATNTYNNNDKANNKDKNKGFSYLNNDIQEDEKNVNKIKKS. 

This protein fragment was used as an antigen for a field-strain-specific capripoxvirus DIVA serological assay.

### 3.2. Confirmation of Protein Fragment Expression, Immunoreactivity, and Purification

The protein fragment expression induced by arabinose and IPTG in BL21 cells had the highest expression after 4 h (Appendix A). Western blot using sheeppox- and goatpox-positive sera showed the reactivity for the fragment with these sera, confirming the immunogenicity (Appendix A). The protein fragment was subjected to purification using the ÄKTA protein purification system. A representative elution graph and SDS-PAGE of the purified product can be seen in Appendix A.

### 3.3. Target Fragment Sequence Confirmation by Mass Spectroscopy

The mass spectrometry analysis of the purified protein fragment confirmed the sequence of the expressed protein (Appendix A).

### 3.4. iELISA Optimization

First, we determined the suitable amount of purified antigen for coating the plates (Appendix A). Then, we optimized the serum dilutions (Appendix A) and the secondary antibody dilutions (Appendix A).

For coating the plates, we found that 60 ng per well of purified antigen (>96%) provided an adequate signal-to-noise ratio. However, due to the cross-reactivity characteristics of the secondary antibody for bovine, sheep, and goat sera, we established two different conditions for the assay, one for sheep and goat (SPP/GTP wts-DIVA iELISA) and another one for cattle (LSD wts-DIVA iELISA). We serially diluted the sera, both cattle and sheep and goat, as well as the secondary antibody. We established that a serum dilution of 1 in 100 produced an adequate response of the positive serum and low background compared to the negative serum for cattle. For sheep and goats, we showed that a serum dilution of 1 in 400 had an excellent response to the positive serum and low background compared to the negative serum. We serially diluted the secondary antibody and established that the dilution of 1 in 30,000 produced the best signal-to-noise ratio for cattle. In contrast, for sheep and goat sera, 1 in 20,000 had the best signal-to-noise ratio.

The LSD wts-DIVA iELISA was tested on well-characterized LSD vaccinated (*n* = 75), naturally infected (*n* = 32), and negative (*n* = 73) serum samples as well as on parapox (*n* = 9) samples for specificity. Additionally, the assay was used on field sera collected during an LSD outbreak in the Republic of North Macedonia (*n* = 45) (Figure 3).

The SPP/GTP wts-DIVA iELISA was tested on well-characterized SPP-vaccinated (*n* = 14), SPP and GTP naturally infected (*n* = 27), and negative (*n* = 87) serum samples as well as on 12 parapox samples for specificity (Figure 4).

For LSD, the capripox wts-DIVA iELISA showed a specificity of 100% and a sensitivity of 99%. For the SPP and GTP field samples tested, the assay showed a specificity of 100% and a sensitivity of 100% and did not detect antibodies in vaccinated or non-specific samples.

### 3.5. Cut-Off, Specificities, and Sensitivities of LSD Wts-DIVA iELISA and SPP/GTP Wts-DIVA iELISA

For the LSD wts-DIVA iELISA, a threshold was obtained by analyzing the data from testing our reference positive and negative sample populations. The cut-off for LSDV was 18.44, determined by ROC analysis and Youden’s index. This result was consistent with the results obtained from the product index. The Euclidean index produced a lower value. We chose the value of 18.44 as it was a higher cut-off and had produced agreement on two of the indexes tested (Appendix A).

A threshold was obtained for the SPP/GTP wts-DIVA iELISA by analyzing the data from testing our reference positive and negative sample populations. As a result, the SPP/GTP wts-DIVA iELISA cut-off was 23.68, determined by ROC analysis and Youden’s index. This result was consistent with the product and Euclidean indexes (Appendix A). 

We applied the above cut-off values to vaccinated or naturally infected positive serum samples tested with the LSD and SPP/GTP wts-DIVA iELISAs. Figure 3 and Figure 4 show that all the naturally infected positive samples are above the cut-off values, while none of the vaccinated ones are.

### 3.6. Cross-Reactivity to Anti-Parapoxvirus Antibodies

Using the above-described protocols, we tested 12 orf-positive goat serum samples, eight pseudocowpox virus positives, and one bovine pustular stomatitis virus-positive serum sample from cattle. Figure 3 and Figure 4 show that all parapoxvirus-positive samples had OD values comparable to cattle-, sheep-, and goat-negative samples.

### 3.7. Antibody Detection in Sera from Longitudinal Studies on Experimentally Infected Animals

Serum samples collected during two independent longitudinal studies with two different viruses were tested using the wts-DIVA iELISA. Appendix A shows that for the two cattle infected with virulent South African LSDV Neethling strain, the seroconversion occurred between 8 DPI and 12 DPI with the highest detection level at 23 DPI. Antibodies were detected at every point after 12 DPI, including the last collected data point at 30 DPI.

Likewise, in two goats infected using GTPV Oman 84, seroconversion occurred between 7 DPI and 14 DPI (Appendix A). The goats remained positive, from 14 DPI to at least 49 DPI.

### 3.8. Practical Application of the Capripox Wts-DIVA iELISA: Investigation of an LSD Outbreak in 2016 in a Vaccinated Herd in the Republic of North Macedonia 

Following the initial outbreak of lumpy skin disease (LSD) in April 2016, North Macedonian veterinary authorities conducted a nationwide mass vaccination campaign using live-attenuated LSD vaccines (Lumpy Skin Disease Vaccine, Onderstepoort Biological Products (OBP) SOC Ltd., SA, and Lumpyvax, MSD, SA). The vaccination program, which commenced on May 25, 2016, targeted all apparently healthy cattle across the country, irrespective of the presence of the virus in the targeted region. Within one to eight weeks (2 to 54 days) after vaccination, multiple holdings nationwide reported clinical observations, including the appearance of skin lesions and increased body temperature among vaccinated animals, prompting further investigations. Under the modified stamping-out policy during this period, only animals displaying clinical symptoms of LSD were culled, with additional sampling conducted for laboratory diagnosis and identification of the virus strain (whether field or vaccinal). Nasal swabs, whole blood, and serum samples were collected from each animal exhibiting LSD-like symptoms for analysis at the laboratory of the Faculty of Veterinary Medicine in Skopje. Nasal swabs and whole blood were primary samples for molecular detection and characterization of LSDV. In contrast, serum samples were archived for future research due to the absence of a commercial ELISA for detecting anti-LSDV antibodies. This study analysed forty-five animal samples, each consisting of paired nasal swabs and serum from the same animals.

After DNA extraction, a duplex real-time qPCR assay for differentiation between virulent and attenuated LSDV strains (DIVA PCR) was conducted on swab samples, revealing that 41 tested positive for the field virus, indicating that these animals were vaccinated while already incubating the disease [45]. One sample tested positive for the vaccine, suggesting vaccine shedding in the nasal swabs of some vaccinated animals. Another sample tested positive for both the vaccine and field viruses, indicating that the vaccinated animal was incubating the virus during vaccination. Two samples tested negative for both the field and vaccine viruses. To retrospectively screen the animals, the wts-DIVA iELISA was used. Of the samples analysed, 43 animals tested positive for antibodies against the wildtype virus, confirming infection with virulent LSDV. Two samples tested negative, suggesting either a delay in seroconversion for infected and vaccinated animals or that the clinical lesions resulted from the vaccination. There was a strong agreement between the DIVA PCR and wts-DIVA iELISA for most samples (39), except for samples 489, 553, 621, 587, 665, and 251. Sample 621 was positive (field) by the DIVA iELISA and vaccine by the DIVA PCR, prompting further analysis through partial sequencing. Additionally, samples 665 and 251, which tested negative in the DIVA PCR, showed positive results for wildtype antibodies in the wts-DIVA iELISA. Samples 489 and 587, positive for the field virus in the DIVA PCR, were negative in the wts-DIVA iELISA but positive in SNT. Lastly, sample 553, positive for both field and vaccine viruses in the PCR, also tested positive in the wts-DIVA iELISA. Information can be found in Appendix A.

A fragment of the B22R gene was sequenced in 39 of these samples to gain more clarity. The results confirmed agreement with wts-DIVA iELISA in 36 samples (92.3%). In the case of sample 621, with discrepancies between the wts-DIVA iELISA and DIVA PCR, a vaccine-like profile was observed, but lacking the A insertion in the B22R gene, typically found in vaccine viruses derived from the Neethling vaccine strain, suggesting a reversion to virulence (Appendix A).

## 4. Discussion

We developed an ELISA capable of detecting antibodies only against wildtype (virulent) LSDV, SPPV, and GTPV in serum samples of cattle, sheep, and goats but not antibodies produced in animals immunized with live-attenuated vaccines.

This DIVA ELISA is based upon altered sequences as a consequence of vaccine-driven viral attenuation of the capripoxvirus homologue of the VARV B22R gene [46,47,48].

Our original design, based on the comparative analysis of the capripoxvirus homologue of the VARV B22R gene for the SPPV vaccine and field viruses, showed the selected region was absent in SPPV vaccines derived from the Romanian and Yugoslavian RM65 strains. Interestingly, this region was also missing in LSDV Neethling-derived vaccines due to an insertion frameshift mutation leading to the truncation of the protein. A similar mutation was also observed in GTPV G20-LKV-derived vaccines.

The sequence alignment indicated that all three target species of capripoxvirus show at least one type of vaccine where the VARV B22R homologue is frameshift mutated, suggesting that this gene could be involved in virulence. In contrast, the targeted fragment was present in all SPPV, GTPV, and LSDV field strains.

This finding was further supported by the domain predictions for capripox field and vaccine viruses, which showed that serum raised against some vaccine strains would not have antibodies against the targeted fragments because the truncation led to the absence of the extracellular domain of the protein containing the target of the current DIVA ELISA.

Our construct generated the intended protein product as confirmed by mass spectroscopy, and its reactivity to capripoxvirus antibodies was confirmed by Western blot using sera raised against field SPPV and GTPV.

Using the expressed product from the fragment of the capripoxvirus homologue of the VARV, we observed clear differences in the reactivity of sera from infected animals (both natural and experimental) and those from vaccinated animals, confirming that this region is altered or absent in the B22R gene product of the vaccine virus. Our data showed positive ELISA results only for animals exposed to capripoxviruses with a wildtype B22R gene profile and not to their vaccine version. Our results indicated that antibodies were raised against the expressed product from the fragment of the capripoxvirus homologue of the VARV B22R, and that was observed across all three capripoxvirus species. The observation that this protein fragment is immunogenic can be supported by an earlier report on orthopoxviruses, which demonstrated the immunogenicity of fragments of the VARV B22R gene product [49]. In fact, the authors used several N-terminal peptides of the monkeypox virus homologue of the VARV B22R virus gene product to develop a peptide ELISA [49]. Indeed, the truncation of the B22R gene has been associated with the attenuation of poxviruses. Legrand et al. (2004) demonstrated that vaccinia virus constructs with a deleted or truncated B22R gene had decreased virulence and replication, resulting in a significant increase in the host’s survival time [50,51]. Previous studies comparing capripoxvirus vaccines and field viruses revealed mutations in the B22R genes of some vaccines within each of the three capripoxvirus species [33].

Since the appearance of lumpy skin disease in Europe and the subsequent expansion to Asia, vaccination with a homologous Neethling-derived live-attenuated vaccine has become the method of choice for controlling the disease. In this circumstance, the availability of a DIVA vaccine is essential for better management of LSD. Indeed, a DIVA strategy is critical to track field virus circulation in a vaccinated population and facilitate the transition from “infected with vaccination” to “disease-free with vaccination” by demonstrating that the antibodies present are not those raised against the wildtype virus. The capripox wts-DIVA iELISA could, for example, rule out infection when a country implements vaccination with a live-attenuated vaccine or when the disease occurs in a vaccinated herd.

To the best of our knowledge, this is the first serological DIVA that detects antibodies against field capripoxvirus, but not the vaccine. Our assay is also the first serological DIVA exploiting naturally occurring changes between virulent and attenuated poxviruses.

Traditionally, DIVA strategies involve deleting at least one antigenic region of a gene encoding for a viral protein in the vaccine virus [20]. The companion test uses this deleted gene product as an antigen to detect antibodies against the field virus but not against the vaccine [10,52]. A second DIVA strategy involves expressing a foreign protein as part of the vaccine. Again, a companion test targets this foreign protein to identify the samples coming from vaccinated individuals [10].

One of the advantages of the capripox wts-DIVA iELISA is that it makes use of the differences between the attenuated and field versions of the current virus strains. Therefore, there is no need for new vaccine development. This is particularly significant since well-established and efficacious capripox vaccines are currently in use and are readily available. Depending upon the circulating viral strain, any currently available vaccine strain with an attenuation resulting in changes of the VARV B22R homologue gene in the region previously indicated could use this assay. This reduces costs for countries interested in applying a DIVA strategy in their control and eradication campaigns.

The approach used here can also be applied to the development of assays for other poxvirus diseases or other DNA viruses when the disease control measure is based on live-attenuated vaccines.

In order to determine vaccine efficacy and establish adequate disease control and eradication programs, the DIVA iELISA should be combined with other serological assays that detect antibodies in both infected and vaccinated animals. The two assays combined can help fully determine a region’s epidemiological status.

A limitation of this assay is that it requires the use of a live-attenuated vaccine containing the described alterations in the B22R gene, such as the LSDV Neethling vaccine, LSDV KSGP 0240 (KS-1), SPPV Romanian strain of SPPV Yugoslavian RM65. Vaccine strains such as SPPV Turkey (MN072631), which does not contain the indicated mutations in the B22R gene, are not expected to work with the DIVA ELISA.

Furthermore, additional testing is required for validation of the assay, such as a study of the DIVA ELISA’s ability to detect field antibodies on DIVA-enabled vaccinated animals after a challenge.

Regarding GTPV vaccines, we did not have any samples available to test. However, based on its design, we expect the capripox wts-DIVA iELISA to detect antibodies against GTPV but not those raised against GTPV G20-LKV or AV41.

We also believe that this assay would not detect antibodies raised against inactivated vaccines as the B22R gene product is a non-structural protein and, therefore, can only be expressed by a replicating virus. Further research is required to address this and similar issues.

The capripox wts-DIVA iELISA will facilitate animal trade-related screening, serosurveillance, outbreak investigation in vaccinated and non-vaccinated herds, declaration of freedom from disease, and specifically, freedom with vaccination. In addition, since this assay exploits common sequence mutations that are found in some of the current attenuated vaccine strains of each capripoxvirus species, it provides a serological DIVA test for those vaccines.

## Figures and Tables

**Figure 1 viruses-15-02318-f001:**
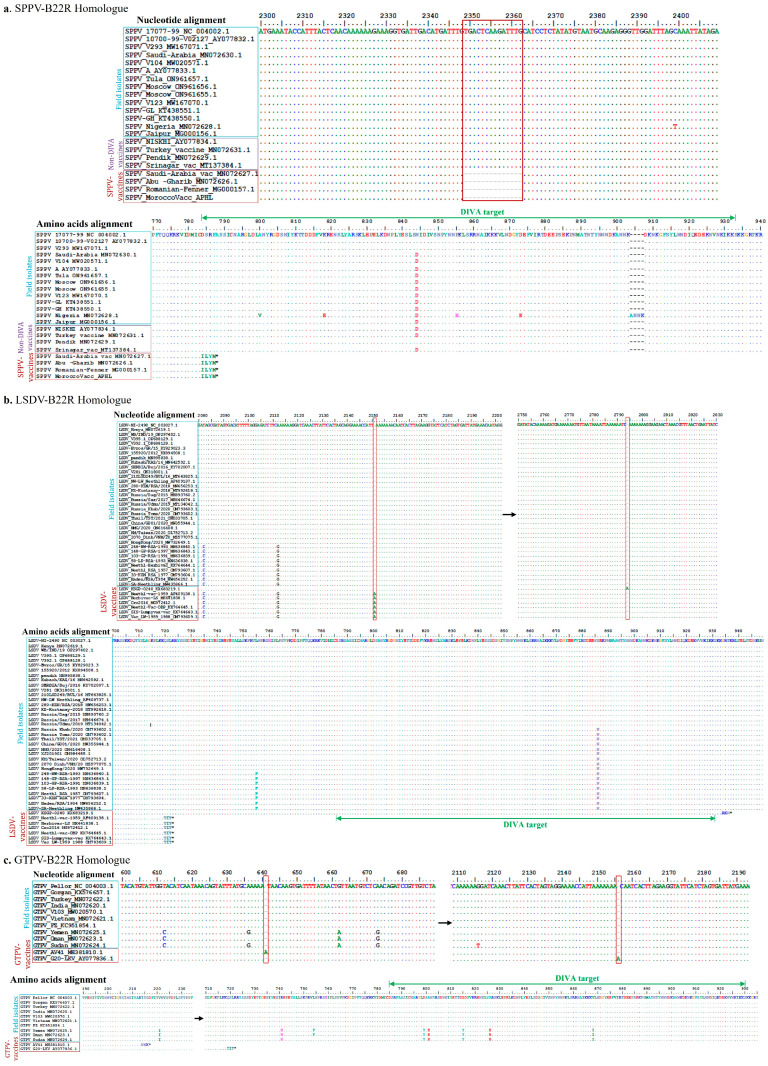
Nucleotide and amino acid alignments of (**a**) SPPV B22, (**b**) LSDV, and (**c**) GTPV homologue genes and their proteins. For SPPV, the nucleotide alignment shows a 14-nucleotide deletion in some of the vaccines but not in the field strains. For LSDV and GTPV, the nucleotide alignments showed an insertion (A) in some of the vaccines but not in the field strains. The mutations resulted in frameshifts and early protein truncations. * SPPV Morocco vaccine—sequenced in-house.

**Figure 2 viruses-15-02318-f002:**
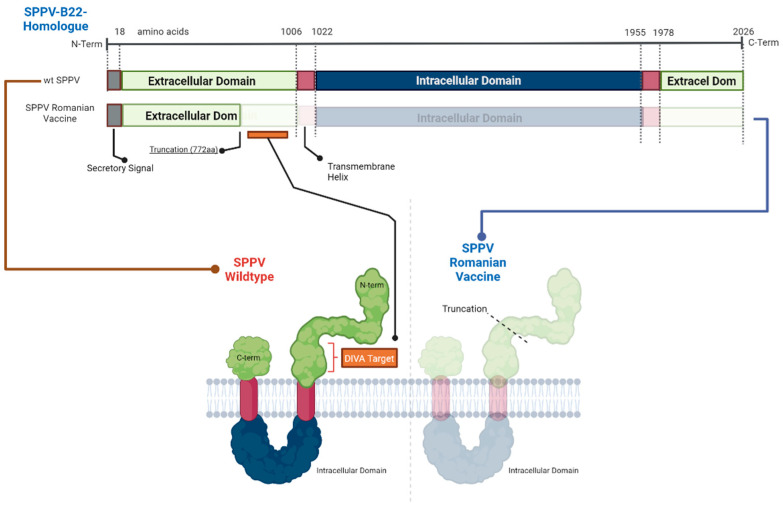
Protein domain predictions of SPPV B22 wildtype and vaccine. Vaccine-related changes would result in a truncated and absent protein. The red square bracket indicates the DIVA target region.

**Figure 3 viruses-15-02318-f003:**
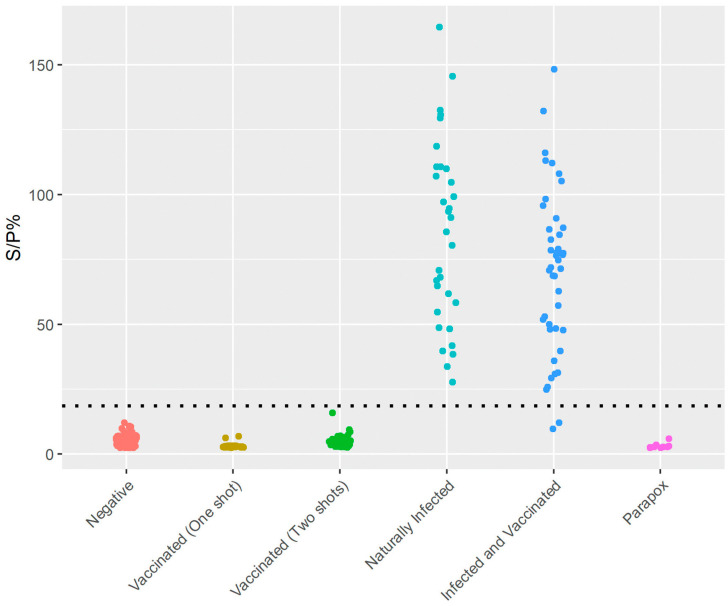
Sample distribution of the LSD wts-DIVA iELISA. LSD negative (*n* = 73), Neethling vaccinated cattle sera (*n* = 75), LSD natural or experimentally infected (*n* = 32), LSD field sera collected during an LSD outbreak in the Republic of North Macedonia (*n* = 45), and pseudocowpox (*n* = 9) serum samples were tested, using conditions for LSD wts-DIVA iELISA. There is a distinct clustering of positive (field) and negative (including vaccine) samples.

**Figure 4 viruses-15-02318-f004:**
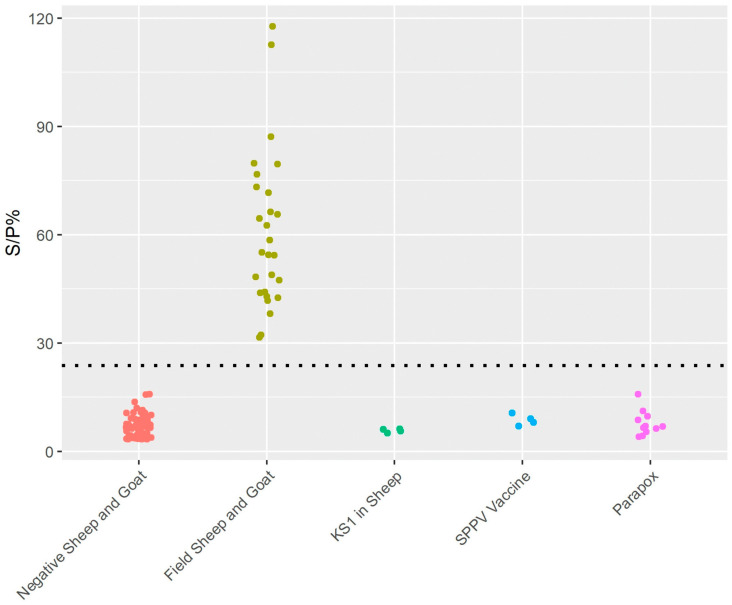
Sample distribution of the SPP/GTP wts-DIVA iELISA. Sheep- or goat-positive (34), SPPV Romanian and LSDV KSGP 0240 (KS-1) sheep-vaccinated sera (8), sheep- and goat-negative (87), and parapox-positive (12) serum samples were tested, using conditions for SPP/GTP wts-DIVA iELISA. None of the vaccinated or non-specific samples were detected.

## Data Availability

The data presented in this study are available upon request from the corresponding author.

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
