# Peer review of "Harnessing Attenuation-Related Mutations of Viral Genomes: Development of a Serological Assay to Differentiate between Capripoxvirus-Infected and -Vaccinated Animals"

_viruses, 2023, doi:10.3390/v15122318_

Round 1

Reviewer 1 Report

Comments and Suggestions for Authors

The authors develop and demonstrate the utility of a serological test to identify infected animals from animals which have been previously vaccinated with several different live attenuated capripoxvirus vaccines or naive unvaccinated animals which have been infected. The principle of this assay is that in the live attenuated vaccines there is a deletion in the variola virus B22R homolog gene which by using the antigen present only in the virulent wild type viruses will allow for vaccinated animals not to elicit antibodies and only antibodies will be elicited in animals which have been infected with field viruses.

This work is novel and most importantly fills a gap in DIVA diagnostics for capripoxviruses.

Minor corrections

Line 213 replace "chessboard" with "checkerboard"

The KSGP-0240 LSDV vaccine does not have the deletion in the B22R homologue.  Displayed in figure 1. Maybe add that this is a non-DIVA vaccine.

The impact of this should be discussed. The KSGP-0249 LSDV vaccine is not fully attenuated and should not be used as a vaccine in cattle and this virus or related virus is circulating in India and surrounding regions. The DIVA assay described would detect antibodies in cattle which have been exposed to this virus.

Figure 4 KS1 vaccinated sheep did not develop antibodies following vaccination.  KS1 is similar to KSGP-0240 and has B22R region used in the ELISA.  Of note the sheep vaccinated with KS-1 did not develop an antibody responses to the B22R region. This can be explained by the low levels of antibody responses in sheep following vaccination with Kenyan (LSDV) vaccines. See Babiuk S, Wallace DB, Smith SJ, Bowden TR, Dalman B, Parkyn G, Copps J, Boyle DB. Detection of antibodies against capripoxviruses using an inactivated sheeppox virus ELISA. Transbound Emerg Dis. 2009 May;56(4):132-41. doi: 10.1111/j.1865-1682.2009.01067.x. Epub 2009 Mar 9. PMID: 19281604. This should be discussed.

Author Response

Answer to Reviewer 1:

a. Line 213 replace "chessboard" with "checkerboard"

Done

b. The KSGP-0240 LSDV vaccine does not have the deletion in the B22R                    homologue.  Displayed in figure 1. Maybe add that this is a non-DIVA vaccine.

See below

c. The impact of this should be discussed. The KSGP-0249 LSDV vaccine is not          fully attenuated and should not be used as a vaccine in cattle and this virus or      related virus is circulating in India and surrounding regions. The DIVA assay          described would detect antibodies in cattle that have been exposed to this            virus.

    Figure 4 KS1 vaccinated sheep did not develop antibodies following                      vaccination.  KS1 is similar to KSGP-0240 and has B22R region used in the              ELISA.  Of note the sheep vaccinated with KS-1 did not develop an antibody          responses to the B22R region. This can be explained by the low levels of                antibody responses in sheep following vaccination with Kenyan (LSDV)                  vaccines. See Babiuk S, Wallace DB, Smith SJ, Bowden TR, Dalman B, Parkyn G,      Copps J, Boyle DB. Detection of antibodies against capripoxviruses using an          inactivated sheeppox virus ELISA. Transbound Emerg Dis. 2009 May;56(4):132-      41. doi: 10.1111/j.1865-1682.2009.01067.x. Epub 2009 Mar 9. PMID: 19281604.      This should be discussed.

This comment and another one from reviewer 2, made us realize that the list of DIVA vaccines in figure 1 included KSGP-0240, but in the text, we called the same vaccine, KS-1.  We corrected this by using the name KSGP-0240 (KS-1) throughout.

We have shown that the KSGP-0240 (KS-1) vaccine is not detected by the DIVA ELISA.  The protein domain prediction and rendering seen in supplementary figure 1, indicate that the B22 protein of KSGP-0240 vaccine is truncated and as a result this protein is not present in host cells.  Additional studies are required to address this and similar issues

Reviewer 2 Report

Comments and Suggestions for Authors

Review of "Harnessing Attenuation-related Mutations of Viral Genomes: Development of a Serological Assay to Differentiate between Capripoxvirus Infected and Vaccinated Animals"

The stated goal of this manuscript is to develop a DIVA ELISA assay with the capacity to distinguish between an antibody response to wild-type Capripoxvirus infected cows, goats or sheep versus an antibody response to Capripoxvirus vaccines to Lumpy Skin Disease virus (LSDV), Sheeppox virus (SPPV) or goatpox virus (GTPV).  To do this, the investigators noted that the B22R protein is homologous in SPPV, GTPV and LSDV and observed (along with others) that selected vaccine strains (not all) of SPPV, GTPV and LSDV have truncation mutations (insertion of single nucleotide leading to frame-shift terminations) in B22R.  Expression in E. coli of a C-terminal fragment of B22R would provide antigen for the detection of antibodies in suspect animals to virulent wild-type/field strain viruses and the lack of antibody to B22R in vaccinated animals.  A serologic based test would be highly useful for serological surveillance, epidemiological studies and useful for capripox eradication efforts.  

Figure 1 is ridiculously small and impossible to read.  

Figure 4- is KS1 sheeppox or lumpy skin disease virus?  Does not mention KS1 in the text.  I assume it is a vaccine strain?  I can't read Figure 1 so I don't know how it is designated.  

The field study results for the LSDV outbreak in Macedonia might have provided an ideal situation for testing the investigators DIVA ELISA assay.  45 samples were analyzed from previously vaccinated cattle (LSDV, Onderstepoort or Lumpyvax vaccines) and the 45 samples were from cattle displaying symptoms of lumpy skin disease.  41 of 45 tested positive for field virus indicating that they were likely infected with LSDV either before or shortly after vaccination.  Unfortunately, this dual exposure to vaccine and field virus makes it difficult to explain the 6 exceptions (out of 45) to their DIVA results.  A more controlled study involving experimental infections with field virus and vaccine virus may be needed to test whether vaccinated cattle that are subsequently exposed (after developing immunity) to field virus develop antibodies to B22R.  This would be highly useful information and would greatly affect the interpretation of their DIVA ELISA test results.  They might mention this in their Discussion. 

A vaccine strain or strains with a synthetic gene (immunogenic) in place of the B22R would also allow detection of vaccinated animals and possibly attenuate at the same time.  It of course would require modifying existing vaccine strains but has the advantage of positively identifying whether an animal has in fact been vaccinated.  Combined with the B22R ELISA, it would give a clear picture of vaccination/exposure status.

Overall, this manuscript provides an important tool for the surveillance and ultimately control (hopefully) of Capripox infections in animals.

Author Response

Answers to Reviewer 2:

a. Figure 1 is ridiculously small and impossible to read.

Issue addressed by considerably increasing the image resolution.

b. Figure 4- is KS1 sheeppox or lumpy skin disease virus?  Does not mention KS1      in the text.  I assume it is a vaccine strain?  I can't read Figure 1 so I don't              know how it is designated.  

This comment as well as one made by Reviewer 1 made us realize that the list of DIVA vaccines included KSGP-0240, but in the text, we called the same vaccine, KS1. 

We corrected this by using the name KSGP-0240 (KS-1) throughout.

c. The field study results for the LSDV outbreak in Macedonia might have                  provided an ideal situation for testing the investigators DIVA ELISA assay.  45        samples were analyzed from previously vaccinated cattle (LSDV,                              Onderstepoort  or Lumpyvax vaccines) and the 45 samples were from cattle          displaying symptoms of lumpy skin disease.  41 of 45 tested positive for field        virus indicating that they were likely infected with LSDV either before or                shortly  after vaccination.  Unfortunately, this dual exposure to vaccine and            field virus makes it difficult to explain the 6 exceptions (out of 45) to their              DIVA  results.  A more controlled study involving experimental infections with        field virus and vaccine virus may be needed to test whether vaccinated cattle        that are subsequently exposed (after developing immunity) to field virus                develop antibodies to B22R.  This would be highly useful information and              would greatly affect the interpretation of their DIVA ELISA test results.  They          might mention this in their Discussion. 

We appreciate this observation and have added it to the discussion.

d. A vaccine strain or strains with a synthetic gene (immunogenic) in place of the      B22R would also allow detection of vaccinated animals and possibly attenuate      at the same time.  It of course would require modifying existing vaccine strains      but has the advantage of positively identifying whether an animal has in fact        been vaccinated.  Combined with the B22R ELISA, it would give a clear picture      of vaccination/exposure status.

Thank you for the comment.  It is well noted.